# On-surface synthesis platform for highly branched oligomers based on sequential C−C coupling and C−H activation of carbenes

Yunjun Cao [1], Joel Mieres-Perez [2], Julien Frederic Rowen [3], Akshay Hemant Raut[3], Paul Schweer [1], Anran Bao[1], Wolfram Sander [3] ✉, Elsa Sanchez-Garcia[2] ✉ & Karina Morgenstern [1] ✉

On-surface synthesis is an emerging field for fabricating low-dimensional nanostructures. While carbenes are versatile reactive intermediates in solution-phase organic synthesis, they have rarely been explored in on-surface synthesis. Here, we demonstrate the versatility of carbenes in synthesizing highly branched zero-dimensional oligomers with distinct structures on a metal surface by combining bond-resolved scanning tunneling microscopy imaging, manipulation, X-ray photoelectron spectroscopy, surface infrared spectroscopy, and ab initio theoretical modeling. We synthesize highly symmetric branched oligomers through the C−C coupling of two carbene molecules to form a core of oligomers, followed by C−H activation of the core with up to four additional carbene molecules to create branches. Branched oligomers of lower symmetry are formed through cyclodehydrogenation of the highly symmetric oligomers. Our on-surface synthetic strategy based on C−H activation of carbene building blocks provides a platform for the design and synthesis of highly branched zero-dimensional oligomers with distinct structures.

On-surface synthesis offers opportunities to fabricate low-dimensional carbon-based nanostructures through various chemical reactions, such as Ullmann coupling[1], Glaser coupling[2,3], and dehydrogenative coupling[4–7]. In recent years, many intriguing nanostructures have been synthesized and characterized on surfaces, ranging from zero-dimensional (0D) fullerenes[8] and cyclocarbons[9] over one-dimensional (1D) polymers[5,10] and graphene nanoribbons[11] to two-dimensional (2D) networks[12]. However, the synthesis of nanostructures gets more challenging with increasing complexity. While generating highly reactive intermediates is essential for the synthesis, controlling these intermediates is necessary to avoid undesired side products and reduced selectivity[13,14]. Developing strategies for a precise and efficient coupling of reactive intermediates is of vital importance for the synthesis of sophisticated low-dimensional nanostructures.

Carbenes are archetypal reactive intermediates of key importance in organic syntheses, undergoing reactions such as dimerization, cyclopropanation, addition to C−C bonds, and insertion into C−H bonds[15–17]. The high reactivity of arylcarbenes (such as diphenylcarbene or fluorenylidene) results from their highly asymmetric electron distribution at the carbene center, resembling a 1,1-zwitter-ionic structure[18]. Despite their versatility in solution chemistry, the use of carbenes in on-surface synthesis has been limited to simple C−C

[1]Physical Chemistry I, Ruhr-Universität Bochum, Bochum, Germany. [2]Computational Bioengineering, Technische Universität Dortmund, Dortmund, Germany. [3]Organic Chemistry II, Ruhr-Universität Bochum, Bochum, Germany. ✉e-mail: wolfram.sander@rub.de; elsa.sanchez@tu-dortmund.de; karina.morgenstern@rub.de

couplings, forming either dimers[19,20] or one-dimensional polymers[21–23]. Carbene insertion into C−H bonds, a promising strategy for C−H functionalization[24], has not yet been integrated into on-surface synthesis protocols.

In this work, we demonstrate the feasibility of carbene-based C−H insertion on a silver surface. Arylcarbenes dimerize via C−C bond formation, and they insert into the C−H bonds of such dimers. Following a controlled reaction sequence of the two reaction steps, highly branched 0D oligomers composed of six carbene units are synthesized (Fig. 1). We characterize the reaction intermediates and the final products of carbenes in detail by bond-resolved scanning tunneling microscopy (STM) imaging and manipulation, X-ray photoelectron spectroscopy (XPS), surface infrared (IR) spectroscopy, and density functional theory (DFT) calculations. Our study expands the on-surface synthesis toolbox, paving the way to synthesize highly branched 0D oligomers with distinct structures using carbenes as building blocks.

## Results and discussion
### On-surface synthesis of carbene 1a

We use 1,8-diaza-9-diazofluorene **1a** as a precursor for the synthesis of 1,8-diazafluorenylidene **2a** on an Ag(111) surface. The surface is beneficial because Ag-catalyzed carbene reactions are useful for various synthetic transformations in organic chemistry, yet they are less explored compared to reactions catalyzed by gold[25]. Studying carbene reactions on well-defined single-crystal silver surfaces provides mechanistic insight into silver-catalyzed carbene reactions in solution. Moreover, Ag surfaces are more promising because of the weaker binding of carbenes to Ag(111) than to Au(111) surfaces[26]. It results from a stronger back-donation from the carbene to the gold surface than to silver at comparable donation from the metal to the carbene[26].

Upon deposition at 50 K, precursor **1a** self-assembles into ribbon patterns (Fig. 2a). Attractive interactions between nitrogen atoms and vicinal hydrogen atoms from a neighboring molecule promote the ribbon assembly via hydrogen bonds (Supplementary Fig. 1).

According to DFT calculations and supported by IR spectroscopy (Supplementary Fig. 2), **1a** is adsorbed with its π-system parallel to the surface (Fig. 2f, bottom). The adsorption geometry is similar to the related 9-diazofluorene **1b**, where the two nitrogen atoms at positions N1 and N8 are replaced by CH groups (Fig. 1). Analysis of the N 1s peak in XPS spectra confirms the non-dissociative adsorption of **1a** at 87 K (Supplementary Fig. 3).

9-Diazofluorene **1b** has been dissociated on the same surface into fluorenylidene **2b** and an $N_2$ molecule by inelastic electron tunneling (IET) manipulation at an electron energy of 1.5 eV[27]. Here, IET manipulations of individual molecules **1a** within the ribbon lead to bright protrusions at a low manipulation voltage of 1.0 V (Fig. 2b–d). These protrusions cannot be converted back to **1a**. More molecules within an extended region around the point of electron injection react to product molecules at a higher manipulation voltage of up to 2.0 V (Supplementary Fig. 4). Based on the IET-induced dissociation of 9-diazofluorene **1b** into fluorenylidene **2b**[27], the product molecules are assigned to carbenes **2a**. The apparent height of **2a** is 70% larger than **1a** at a bias voltage of 50 mV (Fig. 2e). Such a height increase cannot be explained by the charge transfer from the surface, since carbene **2b**, with a similar charge transfer to **2a** (0.5 e vs. 0.6 e), is imaged even less high than its precursor **1b**[27]. It suggests that the product molecules are adsorbed in another geometry than precursor **1a**.

DFT calculations support the identification of the bright protrusions as perpendicularly-adsorbed carbene **2a**. The calculations reveal an adsorption of carbene **2a** via the reactive carbene center with the π-system of the molecule tilted with respect to Ag(111). The angle of 68° between the molecular plane of carbene **2a** and the surface plane contrasts the close-to-parallel geometry of precursor **1a** (Fig. 2f). The highest part of carbene **2a** above the surface plane is, at $h_2 = 574$ pm, around 80% higher than that of precursor **1a** at $h_1 = 311$ pm (Fig. 2f). This change is nicely reflected in the STM images, despite the fact that apparent heights in STM are influenced by both the geometrical heights of adsorbates and their local density of states[28]. The tilted geometry of **2a** is further corroborated by IR spectroscopy based on the surface selection rule[29] (Supplementary Fig. 2).

**Fig. 1 | Reaction pathway of carbenes on Ag(111).** Reaction pathway of precursors **1a** and **1b**, via carbenes **2a** and **2b** towards highly symmetric tetrasubstituted dimers **10a** and **10b**, and less symmetric fused tetrasubstituted dimer **11** on Ag(111). The dashed blue lines indicate the mirror planes.

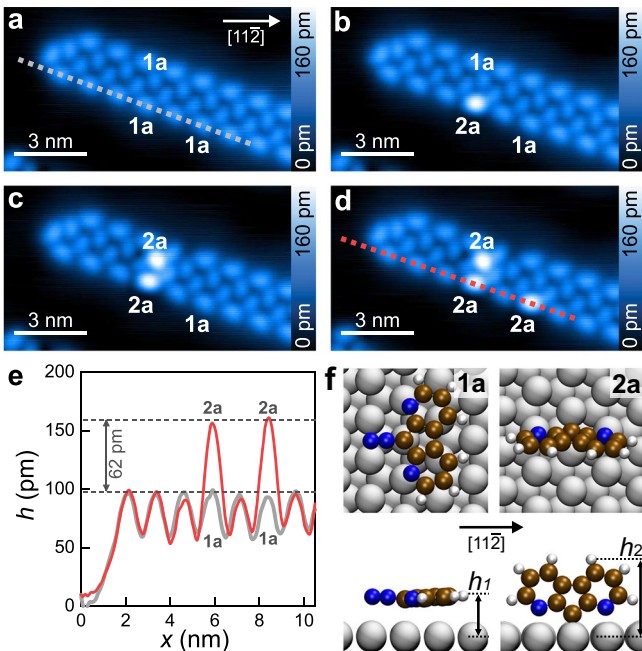

**Fig. 2 | Formation of carbene 2a from precursor 1a. a–d** Sequence of STM images with IET-induced dissociation. The IET manipulations are performed by ramping the bias voltage from 0.01 V to 1.0 V, at a step size of 5 mV and a dwelling time of 0.5 s for each step. Scanning parameters: $V_b = 50$ mV, $I_t = 10$ pA. **e** Apparent height profiles (gray and red curves) along the dashed lines in (**a**) and (**d**), respectively. **f** Optimized geometries (top and side view) of **1a** and **2a** on Ag(111). Gray spheres: silver atoms; brown spheres: carbon atoms; white spheres: hydrogen atoms; blue spheres: nitrogen atoms. $h_1$ and $h_2$ mark the distances from the surface plane to the hydrogen atoms furthest from the surface for **1a** and **2a**, respectively.

The carbene center of **2a** is situated at a bridge site of Ag(111). Its calculated binding energy of 5.12 eV is significantly higher than the 2.95 eV of carbene **2b** on the same surface[27]. The difference between the carbenes **2a** and **2b** is that the backbone of **2a** comprises two nitrogen atoms next to the carbene center (Fig. 1). The enhanced binding energy of carbene **2a** results from the additional interaction of the lone pairs of the two nitrogen atoms with two Ag atoms. The two atoms are at a distance of approximately $\sqrt{3}a$ along an Ag<112> direction, with $a$ the lattice constant of the Ag(111) surface (Fig. 2f, right). An alternative geometry with a planar carbene **2** bonded to an Ag adatom (adsorbed atom) is not possible because there are no adatoms released from the step edges on Ag(111) at the maximum temperature of 52 K during irradiation[30]. For the discussion of alternative adsorption geometries, see Supplementary Note 2.

It is noteworthy that carbenes **2a** and **2b** differ fundamentally from the widely studied N-heterocyclic carbenes (NHCs)[31]. As σ donors, NHCs bind strongly to metal surfaces, leading to self-assembled monolayers with adatom extraction[32–34]. In contrast, carbenes **2a** and **2b** interact with the metal surface both as σ donors and π acceptors with a net electron transfer from the metal to the molecules (0.6 e for **2a** and 0.5 e for **2b**). For an in-depth discussion, see Supplementary Note 4.

### Dimerization of carbene 1a
Having produced carbenes **2a** locally on the surface by IET manipulation, we perform photolysis of **1a** to carbene **2a** on the entire Ag(111) surface. Irradiation at 365 nm for 3 h converts all the precursor molecules of **1a** to **2a**, the same species as obtained by IET manipulation (Fig. 3a, b; for details, see Supplementary Fig. 4). We confirm the dissociation of **1a** by XPS. After irradiation, the N 1$s$ peaks associated with

diazo groups disappear, indicating the desorption of the $N_2$ byproduct (Supplementary Fig. 3). Thus, the use of diazo compounds as precursors provides a residue-free on-surface synthesis route. It is complementary to the widely used Ullmann coupling, which produces halogen byproducts that chemisorb on the surface, hinder the diffusion of reaction intermediates[35,36], and promote alternative reaction pathways[37].

Carbene **2a** aggregates into larger islands during subsequent annealing at temperatures up to 240 K (Supplementary Fig. 5). Other structures form upon annealing at even higher temperatures. Annealing at 292 K leads to chains oriented along the Ag<110> directions. The chains consist of alternating regularly spaced bright ellipsoidal protrusions and dimer-like species. The latter contains two pairs of three protrusions (Fig. 3d, e and Supplementary Fig. 6). Based on their apparent heights, we assign the ellipsoidal bright protrusions to unreacted carbene molecules **2a** (Supplementary Fig. 7). Annealing at a higher temperature of 375 K converts the unreacted carbenes **2a** to six-protrusion species. It increases the abundance of the six-protrusion species from 57.2% to 80.2% (Fig. 3h). In addition, a butterfly-shaped species is synthesized at the higher annealing temperature (Fig. 3g).

The geometry of the six-protrusion species is better visible in one of the rare isolated species on the surface (Fig. 3e). Two rows of bright protrusions are distinguished, each displaying three protrusions, with the middle ones elongated. As the six-protrusion species maintains its integrity during lateral manipulation, its internal bonding is stronger than that of the individual constituents to the surface (Supplementary Fig. 6). We propose that this species is an organometallic intermediate **3a** (Fig. 1), comprising two carbene molecules linked by three Ag adatoms. In an optimized structure, the two diaza-fluorenyl groups adsorb at slightly different angles to the surface plane (Fig. 3j). The different angles in the calculated geometry are reflected in different shapes of the two halves of the isolated organometallic intermediate **3a** in the STM images. The three protrusions on the right side are more elongated than those on the left side (Fig. 3e, i). The shape of the organometallic intermediate **3a** depends on its neighbors. It suggests a geometric flexibility induced by non-directed intermolecular interactions (for details, see Supplementary Notes 7 and 8). Organometallic complexes are typical reaction intermediates in on-surface syntheses[14,38,39].

At a higher annealing temperature of 375 K, some of these intermediates react to a butterfly-shaped species. We assign this species to the metal-free dimer **4a** (cf., Fig. 1). This assignment is confirmed by direct deposition of the dimer **4a** on the Ag(111) surface (see Supplementary Note 9 for its synthesis in wet chemistry). It reveals that the wet-chemistry and on-surface synthesized dimer **4a** are identical. On the surface, the dimer **4a** is formed from the organometallic intermediate **3a** by C−C coupling under the release of the Ag adatoms (Fig. 1). The formation of organometallic intermediates **3a** and dimers **4a** is supported by a splitting of the N 1$s$ peak in the XPS spectrum (Supplementary Fig. 3c).

### C−H activation by carbene 1a
The organometallic intermediate **3a** exhibits a high thermal stability even at 375 K, preventing its complete conversion to the dimer **4a**. It enables a carbene reaction beyond C−C coupling at higher temperatures. Thus, annealing at 600 K converts the dimers **3a** and **4a** into a series of oligomers (Fig. 4a, b(i)−g(i)). Bond-resolved STM images recorded with a functionalized tip[40] confirm these oligomers as substituted dimers **5a** to **10a** (Fig. 4b(ii)−g(ii), b(iii)−g(iii)). Up to four diaza-fluorenyl groups are attached to a dimer **4a** core (Fig. 4b(iv)−(iv)). The covalent bonds between the dimer **4a** core and the diaza-fluorenyl branches are further supported by the structural integrity of these oligomers during lateral manipulations across the surface. It is noteworthy that the C−H activation exhibits high selectivity, attaching exclusively at the C3(3')/C6(6') positions of the dimer **4a** core (Fig. 1).

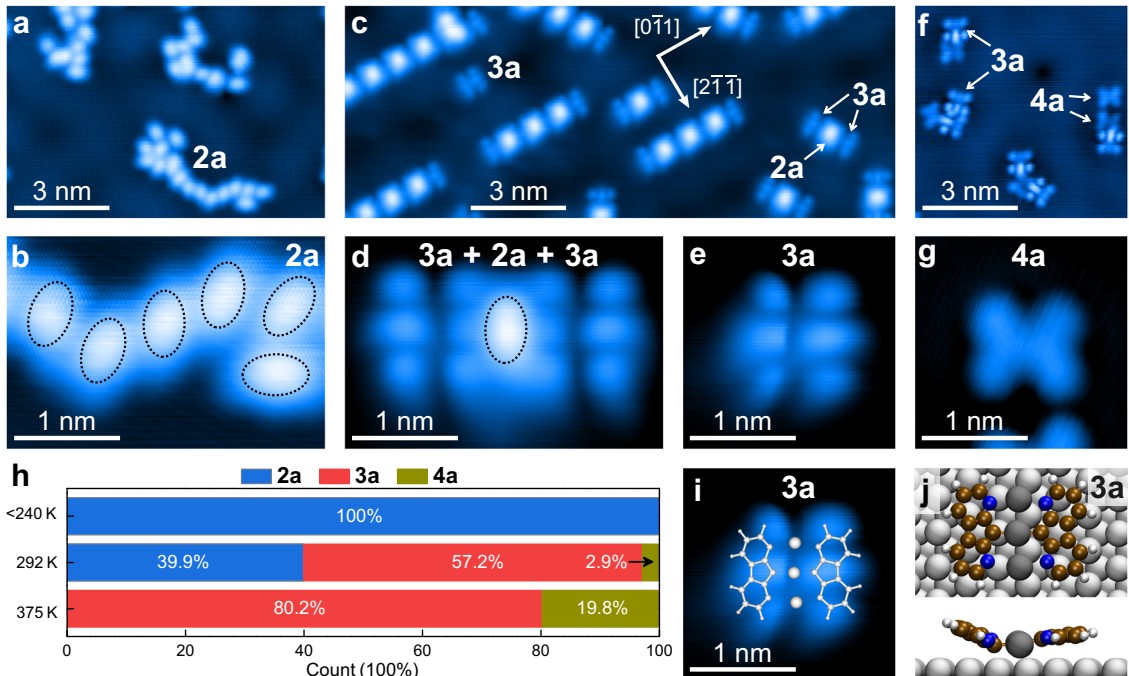

**Fig. 3 | C–C coupling of carbene 2a. a–g** STM images after photolytic dissociation of precursors **1a** at 52 K (**a**, **b**) and subsequent annealing at 292 K (**c–e**) and 375 K (**f**, **g**). Scanning parameters: **a**, **c** $V_b$ = 50 mV, **b**, **d–g** $V_b$ = 10 mV, and **a**, **f**, **g** $I_t$ = 5 pA, **b** $I_t$ = 100 pA, **c** $I_t$ = 10 pA, **d**, **e** $I_t$ = 1 nA, **h** Statistical distribution of carbenes **2a**, organometallic intermediates **3a**, and metal-free dimers **4a** at indicated annealing temperatures. **i** STM image in (**e**) superimposed with an optimized geometry of **3a** on Ag(111). The Ag(111) surface is omitted for clarity. **j** Optimized geometry (top and side view) of organometallic intermediate **3a** on Ag(111). Gray and dark spheres: silver atoms in and above the surface plane, respectively; brown spheres: carbon atoms; white spheres: hydrogen atoms; blue spheres: nitrogen atoms.

Such a site-specific C–H activation is attributed to the steric hindrance between the diaza-fluorenyl branch and the core of dimer **4a**. This steric hindrance only exists under the two-dimensional confinement on the surface[41–43]. It facilitates the synthesis of the 0D oligomers composed of one core and four branches. As carbene **2a** is largely absent on the surface after annealing at 375 K (Figs. 3h), **2a** must be generated by cleavage of the organometallic intermediate **3a** at 600 K. Carbene **2a** formed at such a high temperature readily reacts with dimer **4a** on the surface to form the substituted dimers.

The functionalization of C–H bonds by carbene insertion is a widely used strategy in solution[24]. It proceeds through a concerted process of C–C bond formation and hydrogen atom migration, thereby generating $sp^3$ carbon atoms[44]. In on-surface synthesis of polycyclic aromatic hydrocarbons at elevated temperatures, both dehydrogenation to $sp^2$ carbon atoms[45–47] and preservation of $sp^2$ carbon atoms[48–50] have been reported. To explore whether the $sp^3$ carbon atoms persist in substituted dimers **5a** to **10a**, we perform IET manipulations and confirm the result by d$I$/d$V$ spectra and maps (Supplementary Fig. 11). Substituted dimers **5a** to **10a** contain nondehydrogenated $sp^3$ carbon atoms according to these experiments.

**Intramolecular cyclodehydrogenation**
The branched 0D tetrasubstituted dimer **10a**, formed through C–H activation induced by carbenes, is highly symmetric and exhibits two mirror planes (Fig. 1). To tune the symmetry of such branched structures, we employ intramolecular cyclodehydrogenation, a reaction widely used in on-surface graphene nanoribbon synthesis[11]. However, the pyridine subunits of tetrasubstituted dimer **10a** prevent the two diaza-fluorenyl units in the core from further reaction (Fig. 1). In contrast, such a cyclodehydrogenation reaction is feasible for the tetrasubstituted dimer **10b**. It produces a fused tetrasubstituted dimer **11** with only one mirror plane (Fig. 1).

We synthesize tetrasubstituted dimer **10b** using carbene **2b** as a building block. Following the same procedure used to form carbene

**2a**, we photolyze precursor **1b** using 365 nm irradiation to form carbene **2b** on the Ag(111) surface (Supplementary Fig. 12). The formed carbene **2b** consists of an ellipsoidal protrusion next to a depression. The depression is related to a reduced density of states because the carbene draws electron density from the metal[27].

Upon annealing at 240 K, carbenes **2b** form organometallic intermediates **3b** (Supplementary Fig. 13). It contrasts carbene **2a**, which remains stable at 240 K (Supplementary Fig. 5), and requires annealing at 292 K to form organometallic intermediates **3a** (Supplementary Fig. 6). The bright protrusion in the dimer indicates an Ag adatom connecting two carbenes **2b** (Supplementary Fig. 12). While many molecules of the organometallic intermediates **3b** persist upon annealing at 375 K, some oligomers form. These oligomers consist of bifluorenylidene (dimer **4b**) cores and fluorenyl branches. It indicates C–C coupling and C–H activation of carbene **2b** at 375 K (Supplementary Fig. 13). Annealing at 500 K converts all organometallic intermediates **3b** to oligomers that agglomerate to larger structures (Supplementary Fig. 14). Lateral manipulation to separate these agglomerates identifies substituted dimers of core structures with one to four branches (Fig. 5a–e). It is similar to the substituted dimers formed via C–C coupling and C–H activation of carbene **2a** (Fig. 4b–g). The core structures exhibit two brighter and two less bright protrusions at opposite corners of a tetragon. It is consistent with a twisted geometry of dimer **4b** (Supplementary Fig. 14). Both carbenes follow similar C–C coupling and C–H activation reactions on Ag(111) to form highly symmetric oligomers. However, the reactions proceed at different reaction temperatures because of their distinct surface interactions.

The core structures of the two tetrasubstituted dimers **10a** and **10b** differ (Fig. 1). The dimer **4a** core of **10a** has nitrogen atoms at positions N1(8)/N1'(8'). In contrast, the dimer **4b** core of **10b** has carbon atoms at positions C1(8)/C1'(8'). The twisted structure of the dimer **4b** core facilitates its cyclodehydrogenation at 600 K (Fig. 5f). Upon cyclodehydrogenation on either side, the core structure is planar. It

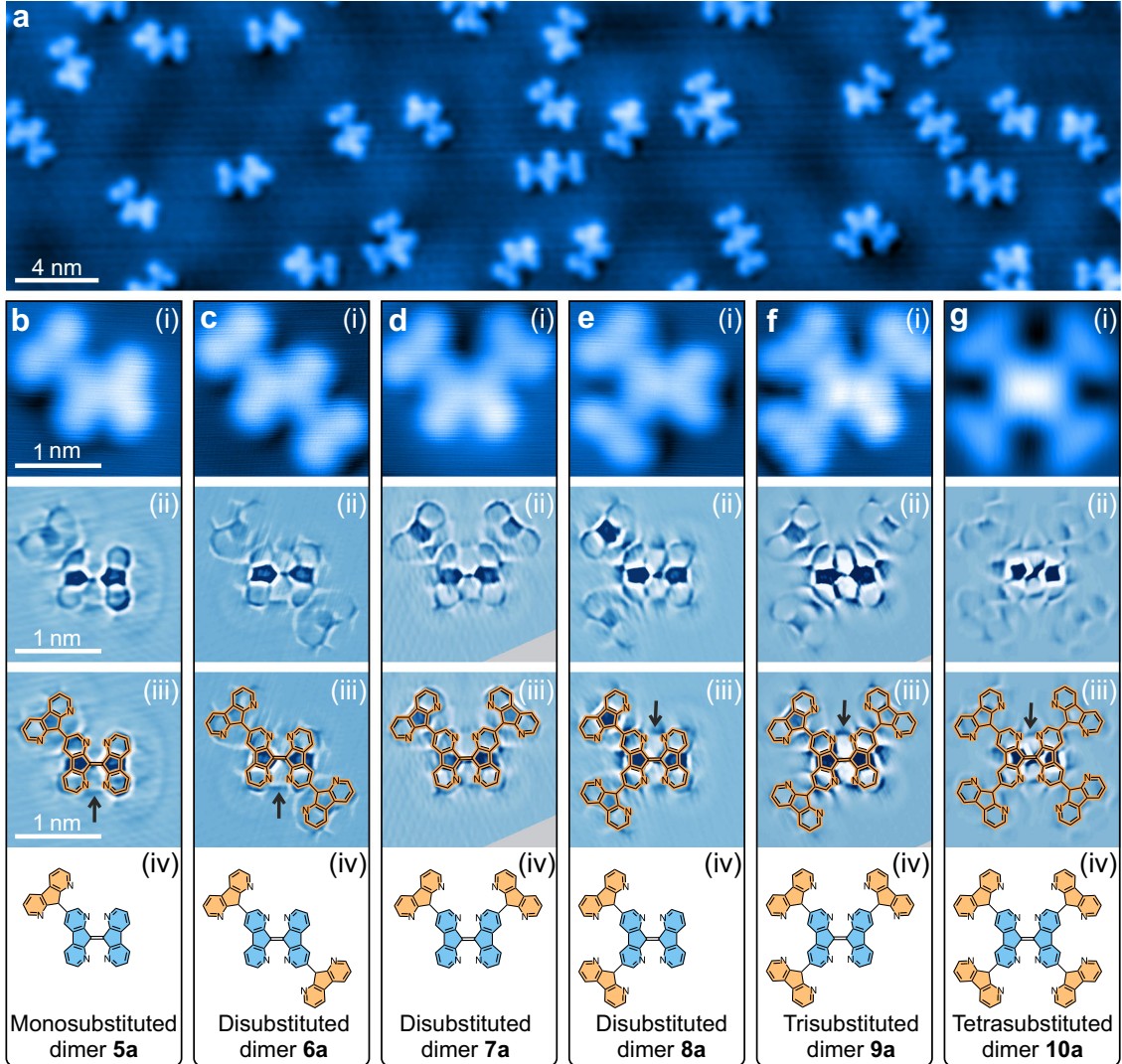

**Fig. 4 | C−H activation of carbene 2a. a** Large-scale STM image after annealing carbenes **2a** on Ag(111) at 600 K. **b**–**g** Small-scale STM images of substituted dimers **5a** to **10a** acquired with (i) a metallic tip in constant-current mode and (ii) a functionalized tip in constant-height mode, respectively; (iii) the constant-height STM images superimposed with chemical drawings for orientation, and (iv) chemical drawings with color filling (core: blue; branches: orange). The images are rotated or mirrored to align the central dimer **4a**. The constant-height images are processed with a Laplace filter. For the original images, see Supplementary Fig. 10. Arrows in (iii) mark imaging artifacts. For details, see Supplementary Note 10. Scanning parameters: **a**, **b**(i)−**g**(i) $V_b = 50$ mV, and **a**, **b**(i)−**f**(i) $I_t = 10$ pA, **g**(i) $I_t = 50$ pA; **b**(ii)−**g**(ii) $V_b = 5$ mV and $I_t = 20$ pA.

prohibits further cyclodehydrogenation on the opposite side. Cyclization at both sides would lead to a non-planar bowl-shaped structure, which is not observed upon annealing at 600 K (for details, see Supplementary Note 14).

Cyclodehydrogenation of asymmetric substituted dimers, such as the monosubstituted dimer **5b** (Fig. 5a), yields two types of fused monosubstituted dimers **12** and **13** (Fig. 5g, h). In contrast, molecules with central inversion symmetry yield only one type of fused tetrasubstituted dimer **11** (Fig. 5i). Tetrasubstituted dimers **10b** and fused tetrasubstituted dimer **11** have distinct structures: **10b** has two mirror planes, while **11** has only one (blues lines in Fig. 1). We generate nine types of fused substituted dimers in two symmetries through cyclodehydrogenation of substituted dimers synthesized from carbenes **2b** (Fig. 5a–e and Supplementary Fig. 15). These dimers either have or lack a mirror plane. Bond-resolved STM images confirm the successful cyclodehydrogenation (Fig. 5g–I and Supplementary Fig. 15). Our results demonstrate that intramolecular cyclodehydrogenation is a feasible approach for tuning the structures of highly branched oligomers on surfaces.

In summary, we have established an on-surface synthesis strategy based on carbenes as building blocks. Highly branched oligomers are synthesized through sequential C−C coupling and C−H activation of the carbenes. The C−C coupling generates dimer cores, which subsequently participate in cross coupling with additional carbenes via C−H activation. The created oligomers contain one to four branches. The success of the synthetic strategy is illustrated using two structurally similar carbenes, one heterocyclic with two pyridine moieties and the other based on a hydrocarbon skeleton. Both carbenes follow the same controlled reaction pathway upon annealing, despite their significantly different surface binding energies (5.12 eV vs. 2.95 eV). An essential ingredient of our strategy is the formation of thermally highly stable organometallic intermediates. Their formation prevents the complete conversion of the two carbenes to C−C coupled dimers. It enables the subsequent C−H activation at elevated temperatures to form symmetrically branched oligomers. Intramolecular cyclodehydrogenation is only feasible for hydrocarbon-based oligomers, leading to less symmetric oligomers.

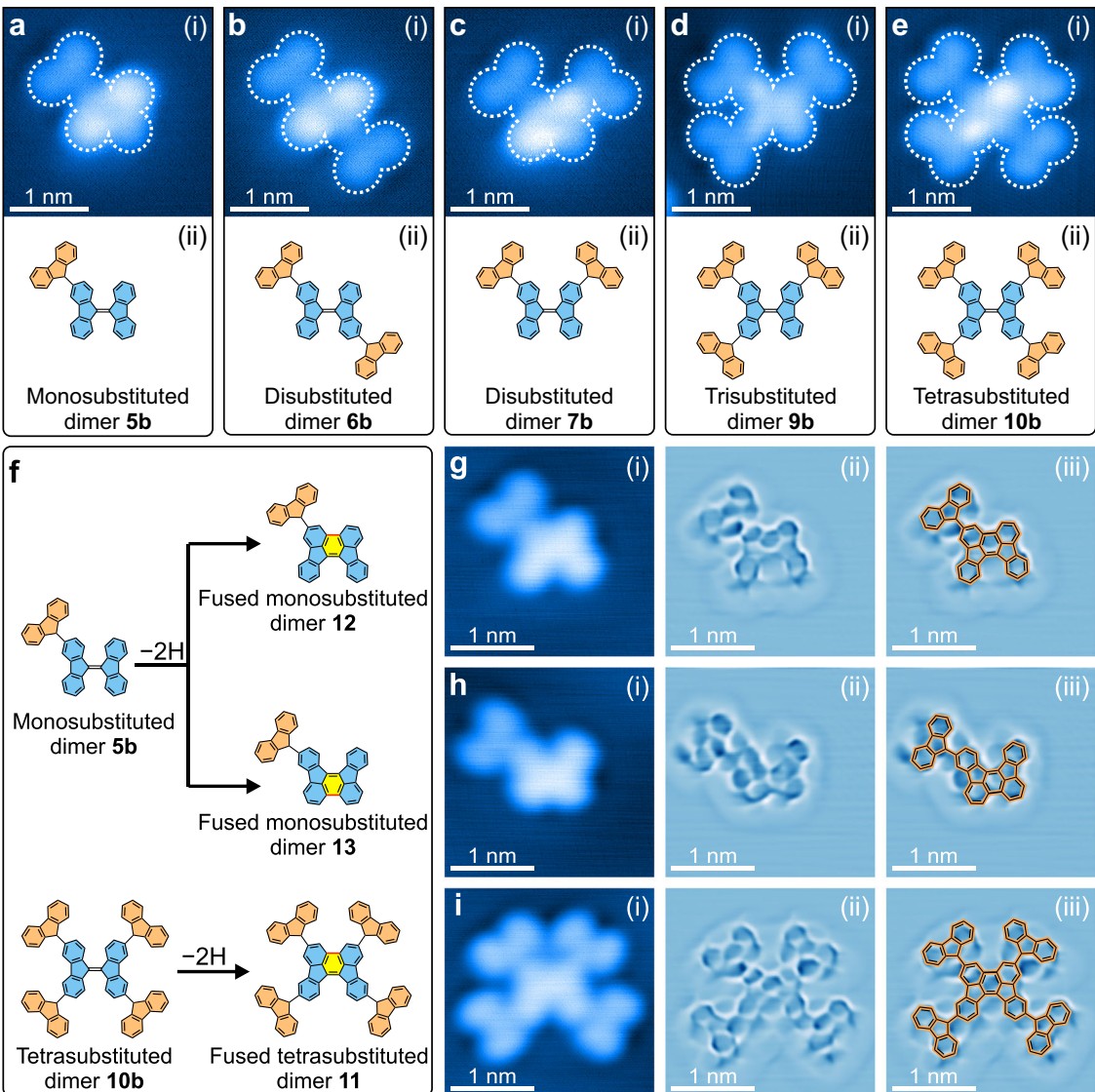

**Fig. 5 | Intermolecular and intramolecular C−H activation. a−e** (i) STM images of substituted dimers after annealing carbenes **2b** on Ag(111) at 500 K, and (ii) corresponding chemical drawings of the substituted dimers. **f** Scheme of the cyclodehydrogenation of monosubstituted dimer **5b** and tetrasubstituted dimer **10b**. **g−i** STM images of the cyclodehydrogenation products acquired with (i) a metallic tip in constant-current mode, (ii) a functionalized tip in constant-height mode, and (iii) constant-height images superimposed with chemical drawings for orientation. The constant-height images are processed with a Laplace filter. For the original images, see Supplementary Fig. 15. Scanning parameters: **a−c** $V_b$ = 300 mV, **d, e** $V_b$ = 200 mV, **g**(i)−**i**(i) $V_b$ = 10 mV, and **a−c** $I_t$ = 2 pA, **d, e** $I_t$ = 3 pA, **g**(i)−**i**(i) $I_t$ = 5 pA; **g**(ii) $V_b$ = 10 mV, $I_t$ = 500 pA, **h**(ii) $V_b$ = 5 mV, $I_t$ = 200 pA, and **i**(ii) $V_b$ = 5 mV, $I_t$ = 300 pA.

Our proof-of-principle study establishes a platform for synthesizing structurally distinct branched oligomers on surfaces. The potential applications of this platform are promising. First, using diazo compounds as precursors of reactive carbenes offers a residue-free on-surface synthesis route that complements the widely used Ullmann coupling on surfaces, thus broadening the toolbox of on-surface synthesis. Second, this approach can be expanded to synthesize various N-doped 0D oligomers by replacing C2, C4, C5, or C7 atoms in carbene **2b** with N atoms. Moreover, the oligomer size can be tuned by tailoring the dimensions of the carbenes. Finally, this approach could be extended to synthesize low-dimensional nanostructures on nonmetallic surfaces. The photo-induced carbene formation does not necessarily require the catalytic effects of a metal surface. The subsequent carbene coupling could be initiated by co-adsorption of single Ag atoms. The platform thus paves the way to synthesize functional materials with applications in nanoelectronics.

## Methods

### Experimental details

The experiments were performed in three UHV systems with similar facilities for sample preparation and molecule deposition. The first system is equipped with a low-temperature STM (Unisoku, base pressure of $1.0 \times 10^{-10}$ mbar), the second with an X-ray photoelectron spectrometer (Specs, base pressure of $4.0 \times 10^{-10}$ mbar), and the third with a vacuum Fourier transform infrared spectroscopy (FTIR) spectrometer (Bruker, VERTEX 80V) coupled to an IR chamber (PREVAC, base pressure of $7.0 \times 10^{-10}$ mbar).

### Sample preparation

The Ag(111) surfaces were cleaned by repeated cycles of 10 min ion sputtering and 10 min annealing at 900 K. In the STM system, the surface was sputtered with 1.0 keV Ne⁺ at $3 \times 10^{-5}$ mbar, yielding an ion current of 1 µA. In the XPS system, the surface was sputtered with 1.3 keV Ar⁺ at $3 \times 10^{-6}$ mbar, yielding an ion current of 12 µA. In the IR

system, the surface was sputtered with 1.0 keV Ar$^+$ at $1 \times 10^{-5}$ mbar, yielding an ion current of 20 µA.

## Molecule deposition

The depositions of precursor **1** and dimer **4a** were performed in molecule deposition chambers separated from the main chambers of the set-up by gate valves. For the synthesis of **1** and **4a**, see Supplementary Note 15. For the deposition of precursor **1**, freshly prepared **1** was transferred in an argon atmosphere into a UHV-cleaned quartz tube sealed by an angle valve, called molecule deposition unit. After loading the molecules, the molecule deposition unit was attached to a molecule deposition chamber, a chamber separated from the preparation chamber of the set-up by a gate valve. The quartz tube was immersed in a cooling bath with an ethanol/LN$_2$ mixture, kept always below 253 K, except during deposition, to avoid thermal dissociation of the precursor. **1** was purified by freeze-pump-thaw cycles; the purity of its vapor was checked by a mass spectrometer. For the deposition of **1a**, the bare Ag(111) surface was placed on a LN$_2$- or LHe-cooled manipulator in the preparation chamber of the STM system. Molecules were dosed through the gate-through valve from the molecule deposition chamber. Thereby, the real pressure at the surface is orders of magnitude lower than the pressure measured in the molecule deposition chamber.

For the STM system, **1a** was deposited on Ag(111) at 50 K for 133 s and 313 s at a pressure of $2.4 \times 10^{-7}$ mbar in the molecule deposition chamber prior to opening the gate-through valve. **1b** was deposited on Ag(111) at 85 K for 163 s at a pressure of $2.1 \times 10^{-7}$ mbar in the molecule deposition chamber prior to opening the gate-through valve. Dimer **4a** was deposited from an organic molecular beam evaporator (OME from MBE Komponenten). The molecule was sublimed at 478 K and deposited on Ag(111) at 140 K for 600 s. After deposition, the samples were transferred to the STM head. The sample temperature increased to 70 K during the transfer.

For the XPS system, **1a** was deposited on Ag(111) at 87 K for 180 s at a pressure of $2.0 \times 10^{-6}$ mbar in the molecule deposition chamber prior to opening the gate-through valve. For the IR system, **1a** was deposited on Ag(111) at 105 K for 300 s at a pressure of $5.0 \times 10^{-7}$ mbar in the molecule deposition chamber prior to opening the gate-through valve. The adsorption temperatures differ slightly due to the different cooling efficiency of the LN$_2$-cooled manipulators. They are well below the thermal dissociation temperature of **1a** and **1b**.

For the STM and XPS system, the photolysis of precursors **1a** and **1b** was induced by a mercury lamp (100 W, Müller GmbH Elektronik-Optik) with a 365 nm bandpass filter (Edmund Optics GmbH) similar to earlier experiments[51]. In the STM system, the Ag(111) surface was in the STM head during irradiation. The sample temperature increased to 52 K during the 3 h irradiation of precursor **1a**, and was below 60 K during stepwise irradiation of precursor **1b** with a total time of 23.2 h. In the XPS system, the sample temperature remained constant at 87 K during irradiation. An irradiation time of one hour corresponds to a photon dose of $2.2 \times 10^{16}$ photons/mm$^2$ for the STM system and $1.0 \times 10^{17}$ photons/mm$^2$ for the XPS system. In the IR system, the photolysis of precursor **1a** was induced with a 365 nm LED light (30 W, Windfire). An irradiation time of one hour corresponds to a photon dose of $4.7 \times 10^{17}$ photons/mm$^2$. The sample temperature remained constant at 105 K during irradiation.

## STM measurements

STM images were recorded with a Pt/Ir tip at 5 K. The bias voltage was applied to the sample. Constant-height STM images were acquired using a functionalized tip with the feedback loop closed at a setpoint on the Ag(111) surface. For direct inelastic electron tunneling (IET) manipulation[28], the tip was positioned above a chosen part of the molecule, and the feedback loop was switched off. A voltage was ramped from 0.01 V to 1.0 V, 2.0 V, or 2.5 V, while recording the

current-voltage (I–V) trace. A steplike change in the tunneling current indicated a successful manipulation. Indirect IET was performed by injecting electrons into the sample at 2 V and 110 pA for 10 s[52]. For lateral manipulation, the STM tip was positioned above a chosen part of the molecule and approached by reducing the tunneling resistance from a typical value of 10 GΩ to 250 kΩ. At this resistance, the tip-molecule interaction is strong enough for manipulation. The STM tip was then moved across the surface along a predefined path in constant-current mode. At its final position, the tip was withdrawn to the imaging distance at the original resistance. The manipulation success was verified in a subsequent STM image.

The dI/dV spectra were recorded in lock-in technique by adding a 20 mV modulation at 413.3 Hz to the bias voltage. The dI/dV maps were recorded at the chosen voltages by recording the lock-in signal at each pixel with the feedback loop on.

The STM topographic images were processed using WSxM[53]. For the Laplace filtering, each STM image pixel is recalculated by multiplying it with the following matrix:

$$M = \begin{pmatrix} 2 & 2 & 1 & 2 & 2 \\ 2 & 0 & -4 & 0 & 2 \\ 1 & -4 & -12 & -4 & 1 \\ 2 & 0 & -4 & 0 & 2 \\ 2 & 2 & 1 & 2 & 2 \end{pmatrix} \quad (1)$$

## XPS measurements

XPS was performed using a Mg K X-ray source with a power of 300 W. The photoelectrons were collected at an angle normal to the surface by a hemispherical analyzer (PHOIBOS 100 from Specs) equipped with a multi-channeltron detector. The binding energy was calibrated at the Ag 3d5/2 peak centered at 368.3 eV. The spectra were analyzed using the commercial software CasaXPS[54].

The molecule peaks in the N 1s spectra were processed before peak fitting. First, a linear background and the satellite peak of the pristine Ag(111) surface was subtracted, with weights based on the corresponding Ag 3d5/2 peaks. Second, the peaks were normalized to the intensity of their corresponding C 1s peaks. Finally, the peaks were fitted with a symmetric Gaussian/ Lorentzian product function.

## IRRAS measurements

The IRRAS measurements were performed in reflection absorption mode at a fixed incidence angle of 80° with a resolution of 4 cm$^{-1}$. The optic bench was evacuated to eliminate absorption from gas-phase species (e.g., H$_2$O and CO$_2$) in the optical path. The infrared reflection absorption spectra were obtained by subtracting a background spectrum recorded before molecule exposure. Each IR spectrum represents an average of 1024 scans.

## Computational details

The calculations were performed with the Quantum-ESPRESSO package[55]. The metal surface was simulated with a silver slab comprising four layers of Ag(111) using (6 × 5) atoms in each layer. For the calculations of species **1a** and **2a**, the bottom two layers were kept fixed at the value of the experimental lattice constant. The rest of the structure was optimized. For species **3a**, the surface atoms (6 × 6 atoms per layer) were kept fixed at the value of the experimental lattice constant. The organic fragments and the three Ag adatoms were allowed to move during the optimization. The atomic simulation environment (ASE)[56] was used to construct the slabs. The PBEsol functional was used for all calculations, employing D3 dispersion corrections to consider the van der Waals interactions[57,58]. Pseudopotentials with a wavefunction cut-off of 50 Ry and a charge density cut-off of 600 Ry were used. The Brillouin zone was sampled using the

gamma point only algorithm, as implemented in Quantum-ESPRESSO. The adsorption distances were calculated as the distance between the adsorbed atoms and the plane formed by the first layer of the surface atoms.

## Data availability

All data are available from the corresponding author upon request. Source data are provided with this paper.

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

## Acknowledgements

This work was supported by the Deutsche Forschungsgemeinschaft (DFG, German Research Foundation) under Germany's Excellence Strategy-EXC-2033-390677874 RESOLV, and by the Research Training group 'Confinement-Controlled Chemistry', which is funded by the Deutsche Forschungsgemeinschaft (DFG, German Research Foundation) under GRK2376//331085229. Y.C. acknowledges the Alexander-von-Humboldt Foundation for a Humboldt Research Fellowship. E.S.G. acknowledges instrumentation funding by the Deutsche Forschungsgemeinschaft (DFG, German Research Foundation) under the large equipment initiative – Project number: 436586093. We acknowledge support by the Open Access Publication Funds of the Ruhr-Universität Bochum.

## Author contributions

K.M., E.S.G., and W.S. supervised the project. Y.C and P.S. performed the STM measurements. Y.C. performed the XPS measurements. Y.C. and A.B. performed the IR measurements. J.F.R and A.H.R. synthesized the precursor molecules. J.M.P. performed the calculations. Y.C. and K.M wrote the manuscript with inputs from all authors.

## Funding

## Competing interests

The authors declare no competing interests.
