## [Transparent Peer Review file · Nature Communications]

On-surface synthesis platform for highly branched oligomers based on sequential C–C coupling and C–H activation of carbenes

Corresponding Author: Professor Karina Morgenstern

Version 0:

Reviewer comments:

Reviewer #1

(Remarks to the Author)

This manuscript is a continuation of the group's previous work on carbenes on metal surfaces. Here, the authors studied the reactions of carbene molecules on Ag(111) where 0D oligomers were formed by sequential C-C coupling and C-H activation, through inelastic electron tunneling (IET) operation, photolysis and thermal control. They also modeled the adsorption configuration using density-functional theory (DFT) calculations. Overall, although this work may expand the scope of carbene chemistry on surfaces, both the significance and data quality do not meet the requirement of Nat. Commun.. In particular, the resolution of BR-STM is not high enough so that the assignment of each structure is not valid. Apart from high-resolution BR-STM and qPlus nc-AFM, spectroscopic techniques like XPS, NEXAFS, IR, or Raman should be performed to elucidate the evolution of reaction intermediates. In addition, the English grammar should be improved. Therefore, I recommend the resubmission of the manuscript in Nat. Commun. or to a more specialized journal (e. g. ACS Nano, Chemical Science) after addressing all the major points above and also scientific issues as listed below.

Major issues :

A five membered ring in a branch of the products (dimers 10, 11) contains a sp³ carbon atom, as shown in Fig. 1 and following associated figures. How can the authors exclude that a H atom is removed during annealing? The BR-STM images rather suggest a planarized adsorption configuration of these products on Ag(111), which implies the sp³ carbon atoms transformed into sp² carbon atoms (a dehydrogenation occurred). This is very common in on-surface chemistry, in particular in the formation of magnetic carbon-based structures (e.g. J. Am. Chem. Soc. 2025, 147, 23, 19530–19538; Angew. Chem. Int. Ed. 2023, 62, e202307884; ACS Nano 2023, 17, 20, 20237–20245).

Once the dehydrogenation reaction occurs, the molecules will probably become open-shell, although the magnetism might not be detected because of the charge transfer from low-work-function Ag substrate to the molecules. However, STS analysis and dI/dV maps of molecular orbitals will offer a guidance, which is thereof suggested to be done.

Minor issues :

1) The structure in Fig. 4g(i) is not clearly indicated in Fig. 4a. The authors should highlight this species in Fig. 4a. Similarly, in Fig. S4, the correspondence between Fig. S4a and Fig. S4f is not clear. It would be advisable to label the relevant structure in Fig. S4a.

2) It was mentioned that the constant height STM image is processed with a Laplace filter. However, this images are significantly distorted as shown in Figs. 4b(ii)-g(ii) and Figs. 5g(ii)-i(ii). I suggest that the authors provide the original images in the Supplementary Information.

3) It seems that some N-N bonds are also formed according to BR-STM images in Fig. 5.

4) In Fig. 5, the authors describe the dehydrogenation of the tetrasubstituted dimer 10b to form the fused tetrasubstituted dimer 11. Theoretically, it is possible to remove four hydrogen atoms within the molecule (rather than just two) to achieve cyclization. The authors should add some discussion to justify this. Did the authors anneal the sample to a higher temperature?

5) While the authors discuss the correlation between reaction temperature and product distribution, I recommend that they also consider the reaction energy barriers and transition states using DFT to rationalize the reaction pathways.

6) English grammar. JUST FOR EXAMPLE.

“Developing strategies for precisely and efficiently coupling reactive intermediates is crucial for synthesizing sophisticated low-dimensional nanostructures”

“It is important to note the fundamental differences between carbenes 2 and the N- heterocyclic carbenes (NHCs) which are

more frequently on surfaces, which are of key importance to understand the surface reactivity of 2.”

Reviewer #2

(Remarks to the Author)

The manuscript entitled: “On-surface synthesis platform for highly branched oligomers based on sequential C–C coupling and C–H activation of carbenes” is a thorough study that employs two strategies for the formation of carbenes. The work presents outstanding data, and the overall presentation is clear. The figures are also clear and self-explanatory. It is a very good work for the area of on-surface synthesis.

However, I have some queries about it:

1. Carbene 2a is formed either by light (Fig. 3) or tip electrons (Fig. 2). The authors claim that the same structure is formed based on a height comparison. However, I have several concerns regarding this claim:

When electrons are involved, the structural model shown in Fig. 2e seems to be quite accurate, as the transformation appears to be local and occurs in the middle of an island. However, when photons are used, the entire island appears to show an increase in height. There are several issues I observe:

1.1. Looking closely at the STM images, after light irradiation, it appears that “2a” is flat rather than nearly vertical. The most likely scenario is that “2a” reacts with an adatom to form a planar structure, similar to the model proposed for “3a.” Have the authors considered a simpler model for the case of light? How does it compare in terms of DFT energy?

1.2. It is intriguing that Figs. 1e and S2c and f show a height increase of exactly 62 pm in all cases. While this value may have an error, the authors should present a height histogram across different images and structures to convince the reader that the increase is indeed the same (within the error bar) in all cases.

1.3. Have the authors considered the possibility that the height increase could be due to electron transfer rather than a structural change?

1.4. The N₂ groups are quite “fragile,” and I wonder whether the mechanism of removal after light-irradiation is correct, or if the increase in temperature caused by 3 hours of irradiation is what actually leads to the removal of these groups. Have the authors attempted thermal removal?

1.5. It is unfortunate that the authors did not follow the reaction using XAS or XPS (at synchrotron), as this could have confirmed some of the proposed mechanisms.

2. The main drawback of the proposed mechanism is the lack of selectivity (see the dispersion of products in Fig. 4a). This is a known issue, particularly in C–H dehydrogenation reactions. Can the carbene-based oligomer formation be directed and induced in a more selective manner, perhaps through a more directional Ullmann coupling? This could improve the selectivity.

3. Similarly to query 1.2, have the authors considered the possibility that in structure “3a,” the final product might involve a surface metal atom rather than an adatom? DFT calculations could be helpful in addressing this.

4. Why did the authors choose Ag? What about Au? It is likely that gold would result in more diffusion, less interaction, and a higher number of adatoms, potentially leading to more complex structures. A brief discussion of how the choice of surface affects the results would be valuable.

5.- Making carbenes into oligomers is important, but doing so on a metal surface reduces the applicability of those oligomers. The article might be much more sound if the authors managed to perform the reaction on an insulating surface. This could work for “2a,” but not for the C–H dehydrogenation, as C–H cleavage typically requires a catalyst. However, the authors could evaporate “adatoms” to create a single-atom catalyst. This is just a speculative idea. Of course, this is not being requested for the manuscript publication.

Reviewer #3

(Remarks to the Author)

This is an elegant study on C-C coupling in carbene molecules, in the context of “on surface synthesis”. Efforts in using surfaces to obtain planar conjugated structures started less than three decades ago and are becoming increasingly popular. This particular investigation focuses on obtaining highly branched oligomers.

The concept and results are original and the study was conducted methodically and rigorously.

Typically the use of adjectives such as “novel” are discouraged, for fairly obvious reasons (novel platform... if this were not novel, we shouldn't even be discussing it).

My main concerns are:

- the references. At this point, forty references for this kind of study is a bare minimum.

- the conclusions are too dry. I encourage the authors to describe a broader outlook, in the form of perspectives for future

work.

Version 1:

Reviewer comments:

Reviewer #1

(Remarks to the Author)

All my previous concerns are satisfactorily addressed after the major revisions. I therefore recommend its publication in Nat. Commun.

Reviewer #2

(Remarks to the Author)

The authors have responded to all my queries, and those of the other reviewers in a satisfactory manner, adding some new information to the manuscript. They have performed IR, and XPS experiments to confirm their models. The manuscript can be accepted for publication. no other comments.

Response letter to reviewers for manuscript NCOMMS-25-37132-T

Comments are in black, replies in blue, and amendments to the manuscript in red.

Reviewer #1

This manuscript is a continuation of the group's previous work on carbenes on metal surfaces. Here, the authors studied the reactions of carbene molecules on Ag(111) where 0D oligomers were formed by sequential C-C coupling and C-H activation, through inelastic electron tunneling (IET) operation, photolysis and thermal control. They also modeled the adsorption configuration using density-functional theory (DFT) calculations. Overall, although this work may expand the scope of carbene chemistry on surfaces, both the significance and data quality do not meet the requirement of Nat. Commun.. In particular, the resolution of BR-STM is not high enough so that the assignment of each structure is not valid. Apart from high-resolution BR-STM and qPlus nc-AFM, spectroscopic techniques like XPS, NEXAFS, IR, or Raman should be performed to elucidate the evolution of reaction intermediates. In addition, the English grammar should be improved. Therefore, I recommend the resubmission of the manuscript in Nat. Commun. or to a more specialized journal (e. g. ACS Nano, Chemical Science) after addressing all the major points above and also scientific issues as listed below.

Reply: We thank the reviewer for the helpful comments and suggestions on improving our manuscript. As recommended, we have performed XPS and IR measurements to identify reaction intermediates chemically (see Figs. R1 and R2 and corresponding discussions). We have also performed IET manipulations to confirm the assignment of the C–H activation products (see Fig. R3 and corresponding discussion). Moreover, we have improved the resolution of the BR-STM images (revised Figs. 4 and 5).

Indeed, few previous studies have investigated the adsorption of carbenes on surfaces, but mostly those of non-reactive N-heterocyclic carbenes. However, the reactivity of reactive carbenes to polycyclic aromatic hydrocarbons, particularly by C–H activation, has not yet been explored. In this work, we demonstrate for the first time carbene-mediated C–H activation on surfaces, which expands the toolbox for on-surface synthesis for the fabrication of sophisticated, low-dimensional nanostructures, such as highly branched oligomers. We have emphasized these results in the revised Abstract and the Introduction. We hope that the additional data and clarifications provided below address the reviewer's concerns.

● XP spectra

Fig. R1. XP spectra on Ag(111). (a,b) XP spectra of the N 1s and C 1s regions on a Ag(111) surface (i), after adsorption of **1a** at 87 K (ii), after irradiation at 365 nm for 2 h (iii), and after annealing at 400 K (iv) and 600 K (v). The black curve in (a) corresponds to a fit of the satellite peak of the Ag(111) surface on a linear background (dashed line). (c) Background-subtracted N 1s spectra from (a) after subtracting the Ag(111) background (curve i in (a)), normalized to the areas of the related C 1s spectra in (b). (d) Structural formulas of the reaction products with N atoms marked in the colors of the fits in (c).

XPS measurements of the N 1s and C 1s regions are performed for the chemical identification of the species on the Ag(111) surface (Fig. R1a,b). A full quantitative analysis is challenging due to the influence of a satellite peak of the pristine Ag(111) surface in the N 1s region (curve i in Fig. R1a) and the coexistence of multiple products with various geometries at elevated temperatures and the partial desorption. Therefore, we only discuss the trends in the proposed reaction mechanism. To analyze the changes in the N 1s spectra, the surface-related satellite peak is subtracted, and the resulting spectra are normalized to the corresponding C 1s peak areas (Fig. R1c).

Deposition of **1a** at 87 K yields three well-separated peaks with an area ratio of approx.1:1:2 (curve ii in Fig. R1c), corresponding to the three types of N atoms in **1a** (Fig. R1d). The area ratio suggests that the peak of higher intensity (light purple) corresponds to pyridinic N atoms within the molecular backbone (Supplementary Ref. 3). The peaks at 403.3 eV (olive) and 401.3 eV (cyan) are assigned to the N atom binding to the C atom in **1a** and the terminal N atom, respectively, based on the N 1s spectrum of **1b** on the Ag(111) surface (Ref. 20 of the revised manuscript). Curve ii thus confirms that **1a** adsorbs non-dissociatively at 87 K.

Upon irradiation at 365 nm, the 403.3 eV (olive) and 401.3 eV (cyan) peaks disappear, confirming that the diazo group is removed from **1a** (curve iii in Fig. R1c). The byproduct, molecular N₂, desorbs at 87 K (Supplementary Ref. 5). The dissociation of **1a** is corroborated by a shift of the C 1s peak from 285.3 eV to 284.9 eV (curves ii to iii in Fig. R1b). A similar red shift was observed for the dissociation of precursor **1b** to carbene **2b** (Ref. 20), suggesting the formation of carbene **2a**. It is accompanied by a shift of the pyridinic N peak from 399.2 eV to 399.1 eV. The broadening of the pyridinic N peak after irradiation reflects the disordered arrangement of **2a** (Supplementary Fig. 4).

The pyridinic N peak splits upon annealing at 400 K (curve iv in Fig. R1c). According to STM results, **3a** and **4a** are formed at this temperature (Fig. 3f). The N atoms in **3a** and **4a** are in different environments. While **3a** contains all N atoms in similar chemical environments, the N atoms of the twisted geometry of **4a** are in two chemical environments, one closer to the surface and the other farther away. Accordingly, the N 1s spectrum is fitted by three peaks: 399.2 eV (orange) for **3a**, and 401.2 eV (blue) and 398.1 eV (green) for **4a**.

Annealing at 600 K converts **3a** and **4a** into a series of oligomers with a **4a** dimer core and diaza-fluorenyl branches (Fig. 4a). Consistently, the two N 1s peaks of **4a** are not altered, while the 399.2 eV peak of **3a** (orange) evolves into a 399.1 eV peak (dark yellow) assigned to the N atoms in the diaza-fluorenyl branches (curve v in Fig. R1c).

The C 1s peak shifts slightly to lower binding energy (Fig. R1b), from 284.9 eV (curve iii) to 284.7 eV (curve v), consistent with the formation of C=C and C–C bonds in the branched oligomers.

Therefore, the XPS results support the reactions deduced in real space. **The XP spectra and their interpretation are added to the Supplementary Information (Supplementary Fig. 3) and referred to in the main text.**

● IR spectra

Fig. R2. IR spectra on Ag(111). (a) IR spectra after adsorption of **1a** at 105 K (i), after irradiation at 365 nm for 5 min (ii), 10 min (iii), 20 min (iv), 30 min (v), and 40 min (vi), and after annealing at 400 K (vii) and 600 K (viii). (b) Scheme of the IR surface selection rule on metal surfaces. (c) Relative intensities of the peaks (A/A_{tot}) marked in (b) vs. irradiation time (t). A and A_{tot} represent the peak areas at time t and at 40 min, respectively.

Besides XPS measurements, we use IR spectroscopy to determine the geometries of the species on the Ag(111) surface. The IR spectrum of precursor **1a** has only a very weak peak at 2084 cm^{-1} (curve i in Fig. R2a), which is assigned to the asymmetric C=N=N stretching mode. It is slightly red-shifted from its value in argon matrices at 2096 cm^{-1} (Supplementary Ref. 1). The low IR intensity for **1a** on the Ag(111) surface is explained by the IR surface selection rule (Ref. 29). On metal surfaces, a vibrational mode is IR active only if there exists a non-vanishing projection of the dynamic dipole moment along the surface normal (Fig. R2b). Thus, the IR spectrum supports the nearly parallel geometry of **1a** on the surface, as predicted by DFT calculations (Fig. 2f).

After irradiation, the formation of carbene **2a** leads to several intense IR peaks (curves ii to vi in Fig. R2a). The peaks at 1611 cm^{-1} and 1171 cm^{-1} are close to the frequencies of the C–C–C bending and asymmetric stretching modes of **2a** in argon matrices, while the peaks at 1369 cm^{-1} , 1320 cm^{-1} , and 1295 cm^{-1} fall within the ranges of the C–C stretching, C–N stretching, and C–H rocking modes. The dominant peak at 1369 cm^{-1} saturates in intensity after 40 min irradiation, indicating complete conversion of precursors **1a** to carbenes **2a** (Fig. R2c). Notably, the 1369 cm^{-1} peak of carbene **2a** is approximately 26 times more intense than the 2084 cm^{-1} peak of precursor **1a**. Such a substantial increase in intensity suggests a large out-of-plane dipole moment component, which suggests an adsorption geometry of **2a** with a large tilt angle. It is consistent with the nearly perpendicular geometry predicted by DFT calculations and the increased apparent height of **2a** relative to **1a** in STM experiments.

The peak shifts after annealing at 400 K (curve vii in Fig. R2a) are consistent with the formation of organometallic dimer **3a**, reflecting a change in bonding environments. After annealing at 600 K, the IR spectrum is almost featureless (curve viii in Fig. R2a). It suggests that the formed oligomers **5a** to **10a** are oriented nearly parallel to the surface, according to the IR surface selection rule.

The IR data clearly demonstrate the geometry change from **1a** to **2a**, as well as the subsequent evolution of **2a** upon annealing. We have included this discussion in the main text and added Fig. R2 in the Supplementary Information as Supplementary Fig. 2.

Major issues:

A five membered ring in a branch of the products (dimers **10**, **11**) contains a sp^3 carbon atom, as shown in Fig. 1 and following associated figures. How can the authors exclude that a H atom is removed during annealing? The BR-STM images rather suggest a planarized adsorption configuration of these products on Ag(111), which implies the sp^3 carbon atoms transformed into sp^2 carbon atoms (a dehydrogenation occurred). This is very common in on-surface chemistry, in particular in the formation of magnetic carbon-based structures (e.g. J. Am. Chem. Soc. 2025, 147, 23, 19530–19538; Angew. Chem. Int. Ed. 2023, 62, e202307884; ACS Nano 2023, 17, 20, 20237–20245).

Once the dehydrogenation reaction occurs, the molecules will probably become open-shell, although the magnetism might not be detected because of the charge transfer from low-work-function Ag substrate to the molecules. However, STS analysis and dI/dV maps of molecular orbitals will offer a guidance, which is thereof suggested to be done.

Reply:

Fig. R3. IET-induced dehydrogenation of sp^3 carbon atoms. (a) Scheme of IET manipulations for **5a** and **10a**. (b-e) STM images of substituted dimers **5a** (b,c) and **10a** (d,e) before and after I - V manipulation up to 2.5 V at the white crosses. (f-i) STM images in (b-e) are displayed with another color scale to enhance the contrast. Scanning parameters: $V_b = 5$ mV, $I_t = 20$ pA. (j) dI/dV spectra recorded at the crosses in (e) in corresponding colors. (k, l) dI/dV maps recorded at 1.02 V (k) and 2.03 V (l) with a tunneling current of 100 pA, a modulation voltage of 20 mV, and a modulation frequency of 413.3 Hz. All images use the same length scale marked in panel (b).

We agree that dehydrogenation of sp^3 carbon atoms is possible at elevated temperatures, as mentioned by the reviewer (Refs. 45-47). However, the persistence of sp^3 carbon atoms at similar temperatures has also been reported in other adsorbed systems (Refs. 48-50).

A generation of sp^3 carbon atoms in our system is consistent with the carbene C-H insertion reaction in solution, which introduces sp^3 carbon atoms through a concerted process of C-C bond formation and hydrogen atom migration (Ref. 44). To confirm whether or not the sp^3

carbon atoms persist, we performed IET manipulations at the carbon of interest and monitored the induced changes using STM/STS and dI/dV mapping (Fig. R3). We expected that a hydrogen can be removed from an sp^3 carbon site in the range of 2 eV to 3 eV (Refs. 45, 50). A similar manipulation at a dehydrogenated sp^2 site should not alter the molecule (Supplementary Ref. 17).

For substituted dimers **5a** and **10a**, the manipulations clearly alter the molecules (Figs. R3b-f). The adjacent depressions in the subsequent STM images after manipulation (Figs. R3g, i) indicate a charge transfer from the molecule to the surface (Ref. 28), consistent with the conversion from an sp^3 to an sp^2 carbon atom. For **5a**, one corner of the dimer becomes brighter, most likely due to a tilting of the pyridine ring (gray arrows in Fig. R3a and Fig. 3c). It can be caused by a reduced molecule–surface distance at the newly formed sp^2 carbon site. For **10a**, a similar topographic change is quenched by a diaza-fluorenyl group at the same position. Therefore, we performed dI/dV spectroscopy and dI/dV mapping to monitor the electronic states of **10a** after manipulation. The spectra show two unoccupied molecular states at 1.02 V and 2.03 V at the dimer and the manipulated part of the molecule, respectively (Fig. R3j). The dI/dV map at 1.02 V is dominated by a symmetric protrusion at the core of the molecule (Fig. R3k). In contrast, in the dI/dV map at 2.03 V the C_2 symmetry is broken and the local density of states at the manipulation site is reduced (red arrow in Fig. R3l). The neighboring branch is brighter (gray arrow in Fig. R3l), likely due to a geometry change. No Kondo resonance is observed in the dI/dV spectra, which is consistent with the quenched magnetism caused by the charge transfer, as suggested by the reviewer.

In summary, the site-specific manipulation, the characteristic manipulation voltages (< 3 V), and the correlated topographic and electronic changes support the presence of sp^3 carbon atoms in the substituted dimers. We have incorporated this discussion into the revised manuscript (page 11) and included Fig. R3 as Supplementary Fig. 11.

Minor issues:

1) The structure in Fig. 4g(i) is not clearly indicated in Fig. 4a. The authors should highlight this species in Fig. 4a. Similarly, in Fig. S4, the correspondence between Fig. S4a and Fig. S4f is not clear. It would be advisable to label the relevant structure in Fig. S4a.

Reply: We are sorry for the misleading wording. There is no correspondence between Fig. 4g(i) and Fig. 4a, or between Supplementary Fig. 6a (former Fig. S4a) and Supplementary Fig. 6f (former Fig. S4f), as the small-scale images are from other overview images. We have revised the figure captions to clarify this point.

2) It was mentioned that the constant height STM image is processed with a Laplace filter. However, these images are significantly distorted as shown in Figs. 4b(ii)–g(ii) and Figs. 5g(ii)–i(ii). I suggest that the authors provide the original images in the Supplementary Information.

Reply:

Fig. R4. Bond-resolved STM images of substituted dimers 5a to 10a. Top row: original constant-height STM images recorded with a functionalized tip; bottom row: corresponding images processed with a Laplace filter. The images are rotated or mirrored to align the central dimer 4a. Scanning parameters: feedback closed with a setpoint of $V_b = 5$ mV and $I_t = 20$ pA.

We thank the reviewer for the suggestion. The original constant-height STM images corresponding to Figs. 4b(ii)-g(ii) have been added as Supplementary Fig. 10 (Fig. R4), and those corresponding to Figs. 5g(ii)-i(ii) are now included in Supplementary Fig. 15.

3) It seems that some N–N bonds are also formed according to BR-STM images in Fig. 5.

Reply:

Fig. R5. AFM imaging and simulations of a tetramer of bis(para-pyridyl)acetylene molecules. Imaging with a CO tip leads to a line between the nitrogen atoms, which could be an image artifact falsely interpreted as a bond (red arrow in (e), adapted from Supplementary Ref. 13).

In Fig. 5, only C–C bonds are formed. The reviewer may be referring to the lines (black arrows) in Fig. 4. These lines are well-known imaging artifacts in high-resolution AFM/STM images, arising from bending of the probe molecule on the tip between two closely-spaced atoms adjacent molecules (Supplementary Refs. 12,13). Such artifacts appear between adjacent

atoms that do not bond (e.g., red arrow in Fig. R5e). We have marked these artifacts in Fig. 4 and clarified that they are not real bonds in the figure caption (page 10).

4) In Fig. 5, the authors describe the dehydrogenation of the tetrasubstituted dimer **10b** to form the fused tetrasubstituted dimer **11**. Theoretically, it is possible to remove four hydrogen atoms within the molecule (rather than just two) to achieve cyclization. The authors should add some discussion to justify this. Did the authors anneal the sample to a higher temperature?

Reply:

Fig. R6. Intramolecular and intermolecular dehydrogenation. (a) Ball-and-stick model of C_{60} fullerene, with five- and six-membered rings numbered in the yellow region. (b) Large-scale STM image after annealing carbenes **2a** on Ag(111) at 700 K. (c) Zoomed-in image of the square in (b). Scanning parameters: (b) $V_b = 50$ mV, $I_t = 10$ pA, (c) $V_b = 10$ mV, $I_t = 1$ nA.

Cyclization of **10b** by removing four hydrogen atoms would lead to a non-planar bowl-shaped structure, similar to a fragment of a C_{60} molecule (yellow region in Fig. R6a). Such a structure is energetically less favorable compared to the planar molecules (Supplementary Ref. 19). At the annealing temperature of 600 K, we did not observe the formation of non-planar molecules indicative of the removal of four hydrogen atoms. We did not explore higher temperatures because annealing **10a** at higher temperatures leads to disordered polymeric structures (Fig. R6b,c). We have added the discussion in the revised manuscript (page 14) and the Supplementary Information.

5) While the authors discuss the correlation between reaction temperature and product distribution, I recommend that they also consider the reaction energy barriers and transition states using DFT to rationalize the reaction pathways.

Reply: We appreciate the reviewer's suggestion. However, it will not be possible to explore all steps of the reaction fully without an automated quantum chemistry workflow, as is already necessary for much smaller systems (e.g., J. Phys. Chem. C 2025, 129, 3469). Though it is nowadays very popular to add some reaction energy scheme, it is unlikely that these present the lowest energy pathways, as frequencies stated in the studies (ACS Catal. 2024, 14, 14206; J. Phys. Chem. C 2009, 113, 710). Thus, it is not so obvious what scientific insight they provide.

Particularly for our work, we consider the further insight we provide by the XPS and IR measurements performed for this reply and presented above to be much more valuable.

6) English grammar. JUST FOR EXAMPLE.

“Developing strategies for precisely and efficiently coupling reactive intermediates is crucial for synthesizing sophisticated low-dimensional nanostructures”

“It is important to note the fundamental differences between carbenes **2** and the N-heterocyclic carbenes (NHCs) which are more frequently on surfaces, which are of key importance to understand the surface reactivity of **2**.”

Reply: We apologize for the grammatical issues. We have made a thorough proofreading of the manuscript and the Supplementary Information, and additionally checked the text with the program Grammarly. All corrections are highlighted in yellow.

With the additional XPS and IR measurements, IET manipulation, and the above clarifications, we hope that we have removed the reviewer’s concerns fully.

Reviewer #2

The manuscript entitled: “On-surface synthesis platform for highly branched oligomers based on sequential C–C coupling and C–H activation of carbenes” is a thorough study that employs two strategies for the formation of carbenes.

The work presents outstanding data, and the overall presentation is clear. The figures are also clear and self-explanatory. It is a very good work for the area of on-surface synthesis.

Reply: We thank the reviewer for naming our study “outstanding” and as “a very good work for the area of on-surface synthesis”. The points raised by the reviewer are carefully addressed as detailed below.

However, I have some queries about it:

1. Carbene **2a** is formed either by light (Fig. 3) or tip electrons (Fig. 2). The authors claim that the same structure is formed based on a height comparison. However, I have several concerns regarding this claim:

When electrons are involved, the structural model shown in Fig. 2e seems to be quite accurate, as the transformation appears to be local and occurs in the middle of an island. However, when photons are used, the entire island appears to show an increase in height.

Reply:

Fig. R7. Incomplete conversion of precursors 1a to carbenes 2a.

(a) STM images after irradiation of precursors **1a** on Ag(111) at 365 nm for 2 h. (b) Apparent height profile along the line in (a). Scanning parameters: $V_b = 50$ mV, $I_t = 10$ pA.

In the original manuscript, we presented only the complete photo-induced conversion of precursors **1a** to carbenes **2a**. We have now added data for an intermediate step with incomplete conversion (Fig. R7a). The height profile clearly reveals that **2a** has a larger apparent height than **1a** in the same STM image (Fig. R7b).

There are several issues I observe:

1.1. Looking closely at the STM images, after light irradiation, it appears that “**2a**” is flat rather than nearly vertical. The most likely scenario is that “**2a**” reacts with an adatom to form a planar structure, similar to the model proposed for “**3a**”. Have the authors considered a simpler model for the case of light? How does it compare in terms of DFT energy?

Reply:

Fig. R8. IR spectra of **1a and **2a** on Ag(111).** (a) Scheme of the IR surface selection rule on metal surfaces. (b) IR spectra after adsorption of **1a** on the Ag(111) surface at 105 K (i), and after irradiation at 365 nm for 5 min (ii), 10 min (iii), 20 min (iv), 30 min (v), and 40 min (vi). (c) Relative intensities of the peaks (A/A_{tot}) marked in (b) vs. irradiation time (t). A and A_{tot} represent the peak areas at time t and at 40 min, respectively.

We thank the reviewer for the careful analysis of the STM images and the suggestion of an alternative planar geometry of carbene **2a**. We rule out this possibility based on the IR measurements. According to the IR surface selection rule on metal surfaces (Ref. 29 of the revised manuscript), a vibrational mode is IR active only if the dynamic dipole moment has a non-zero projection along the surface normal (Fig. R8a). Consequently, precursor **1a** that adsorbs nearly parallel to the surface (Fig. 2f), shows a very weak IR peak at 2084 cm^{-1} (curve i in Fig. R8b). It contrasts a strong asymmetric C=N=N stretching peak at 2096 cm^{-1} of **1a** in argon matrices (Supplementary Ref. 1). Upon irradiation, the formation of carbene **2a** leads to several intense IR peaks (curves ii to vi in Fig. R8b). The dominant peak at 1369 cm^{-1} saturates in intensity after 40 min irradiation, indicating complete conversion of precursors **1a** to carbenes **2a** (Fig. R8c). Notably, the 1369 cm^{-1} peak of carbene **2a** is approximately 26 times more intense than the 2084 cm^{-1} peak of precursor **1a**. Such a substantial increase in intensity suggests a large out-of-plane dipole moment component, which can only be explained by a large tilt angle of **2a** relative to the surface.

Moreover, height profiles show the same apparent height increase of $\sim 70\%$ towards precursor **1a** for both photo-induced and IET-induced carbenes **2a** (Supplementary Fig. 4). A height pixel

histogram corroborates the similarity between the two experimental results (see below). Such a substantial height increase is not consistent with a planar adsorption geometry, in particular in view of the reactive carbon atom being drawn closer to the surface (Ref. 27).

In addition, during light irradiation, the maximum surface temperature remains below 52 K (5 K for IET manipulation). At these temperatures, no adatoms are released from the step edges on the Ag(111) surface (Ref. 30). Therefore, the model “2a” with an adatom is ruled out.

We have clarified this point in the revised manuscript and included the IR data in the Supplementary Information.

1.2. It is intriguing that Figs. 1e and S2c and f show a height increase of exactly 62 pm in all cases. While this value may have an error, the authors should present a height histogram across different images and structures to convince the reader that the increase is indeed the same (within the error bar) in all cases.

Reply:

Fig. R9. IET-induced vs. photolytic dissociation of precursor 1a to carbene 2a. (a, b) STM images before (a) and after (b) indirect IET manipulation at the white cross. Manipulation parameters: $V_b = 2.0$ V, $I_t = 105$ pA, and $t = 10$ s. White circles mark the primary region of the dissociation events to guide the eye. (c, d) STM images after irradiation of 1a at 365 nm for 2 h (c) and 3 h (d). Scanning parameters: $V_b = 50$ mV, and (a,b) $I_t = 5$ pA, (d,e) 10 pA. (e, f) Apparent height profiles along the lines in (a) to (d). (g) Pixel histograms of apparent heights normalized to the total number of pixels (N_{tot}) with Gaussian fits. The fitted peak positions are $h_1 = (68 \pm 1)$ pm, $h_2 = (144 \pm 1)$ pm, and $h_3 = (147 \pm 1)$ pm. The surface baselines are set to 0 pm.

We thank the reviewer for bringing this point to our attention. The pixel histograms of apparent heights for precursor 1a (gray), IET-induced carbene 2a (red), and photo-induced carbene 2a (cyan) each are based on statistics from approximately 1000 molecules (Fig. R9a). The peak-

to-peak height difference from **1a** to IET-induced **2a** is (62 ± 7) pm, and to photo-induced **2a** it is (68 ± 9) pm. The two mean values are identical within the error bar. Moreover, height profiles across some molecules show height increases of (76 ± 1) pm for IET-induced **2a** and (79 ± 1) pm for photo-induced **2a** (Fig. R9b), also identical within the error range.

It should be noted that pixel histograms and height profiles measure apparent heights differently. Histograms capture the most frequently occurring apparent heights in the STM image, while height profiles measure the maximum apparent heights along the lines. Thus, the absolute apparent height values are bigger when determined from height profiles than from pixel histograms, and the relative height values are smaller. Nonetheless, both analyses confirm that the IET-induced and photo-induced **2a** have the same apparent height and are significantly higher than that of **1a**. We have added the error analysis in the Supplementary Information and incorporated Fig. R9 as Supplementary Fig. 4.

1.3. Have the authors considered the possibility that the height increase could be due to electron transfer rather than a structural change?

Reply: We agree with the reviewer that charge transfer influences apparent heights. In our case, **2b** has a similar charge transfer to **2a** (0.5 e vs. 0.6 e), but it is imaged at an even lower apparent height than its precursor **1b** (Supplementary Fig. 12). Therefore, charge transfer cannot explain the substantial height increase for **2a** compared to **1a**. We have clarified this point in the revised manuscript (page 5).

1.4. The N_2 groups are quite “fragile,” and I wonder whether the mechanism of removal after light-irradiation is correct, or if the increase in temperature caused by 3 hours of irradiation is what actually leads to the removal of these groups. Have the authors attempted thermal removal?

Reply:

Fig. R10. N 1s XP spectrum of **1a** on Ag(111) with fits. The N atoms in the structural formula are highlighted in the colors of the fits. The pristine Ag(111) satellite peak is subtracted for clarity.

We thank the reviewer for raising this point. The irradiation temperatures for **1a** (below 52 K) and **1b** (below 60 K) are lower than their respective deposition temperatures (**1a** at 70 K and **1b** at 85 K).

XPS confirms that the molecule remains intact upon adsorption (Fig. R10). Deposition of **1a** at 87 K yields three well-separated peaks with an area ratio of approx. 1:1:2, corresponding to the three types of N atoms in **1a** (Fig. R10). The area ratio suggests that the peak of higher intensity (light purple) corresponds to pyridinic N atoms within the molecular backbone (Supplementary Ref. 3). Based on the N 1s spectrum of **1b** on Ag(111) (Ref. 20), the peaks at 403.3 eV (olive) and 401.3 eV (cyan) are assigned to the N atom binding to the C atom in **1a** and the terminal N atom, respectively. These results confirm that **1a** adsorbs non-dissociatively at 87 K, well above the maximum temperature of 52 K during irradiation. Thus, the temperature increase during irradiation is not sufficient to remove the diazo groups. We have added this discussion to the revised manuscript and incorporated Fig. R10 to Supplementary Fig. 3.

1.5. It is unfortunate that the authors did not follow the reaction using XAS or XPS (at synchrotron), as this could have confirmed some of the proposed mechanisms.

Reply:

Fig. R11. XP spectra on Ag(111). (a,b) XP spectra of the N 1s and C 1s regions on a Ag(111) surface (i), after adsorption of **1a** at 87 K (ii), after irradiation at 365 nm for 2 h (iii), and after annealing at 400 K (iv) and 600 K (v). The black curve in (a) corresponds to a fit of the satellite peak of the Ag(111) surface on a linear background (dashed line). (c) Background-subtracted N 1s spectra from (a) after subtracting the Ag(111) background (curve i in (a)), normalized to the areas of the related C 1s spectra in (b). (d) Structural formulas of the reaction products with N atoms marked in the colors of the fits in (c).

We have performed XPS measurements to support the STM observations (Fig. R11). To analyze the changes in the N 1s spectra, the surface-related satellite peak is subtracted, and the resulting spectra are normalized to the corresponding C 1s peak areas (Fig. R11c).

As pointed out in the reply to the previous question, the three well-separated peaks for **1a** confirm non-dissociative adsorption at 87 K (Fig. R11c,d). Upon irradiation at 365 nm, the 403.3 eV (olive) and 401.3 eV (cyan) peaks disappear, confirming that the diazo group is removed from **1a** (curve iii in Fig. R11c). The byproduct, molecular N₂, desorbs at 87 K

(Supplementary Ref. 5). The dissociation of **1a** is corroborated by a shift of the C 1s peak from 285.3 eV to 284.9 eV (curves ii to iii in Fig. R11b). A similar red shift was observed for the dissociation of precursor **1b** to carbene **2b** (Ref. 20), suggesting the formation of carbene **2a**. It is accompanied by a shift of the pyridinic N peak from 399.2 eV to 399.1 eV. The broadening of the pyridinic N peak after irradiation reflects the disordered arrangement of **2a** (Supplementary Fig. 4).

The pyridinic N peak splits upon annealing at 400 K (curve iv in Fig. R11c). According to STM results, **3a** and **4a** are formed at this temperature (Fig. 3f). The N atoms in **3a** and **4a** are in different environments. While **3a** contains all N atoms in similar chemical environments, N atoms of the twisted geometry of **4a** are in two chemical environments, one closer to the surface and the other farther away. Accordingly, the N 1s spectrum is fitted by three peaks: 399.2 eV (orange) for **3a**, and 401.2 eV (blue) and 398.1 eV (green) for **4a**.

Annealing at 600 K converts **3a** and **4a** into a series of oligomers with a **4a** dimer core and diaza-fluorenyl branches (Fig. 4a). Consistently, the two N 1s peaks of **4a** are not altered, while the 399.2 eV peak of **3a** (orange) evolves into a 399.1 eV peak (dark yellow) assigned to the N atoms in the diaza-fluorenyl branches (curve v in Fig. R11c).

The C 1s peak shifts slightly to lower binding energy (Fig. R11b), from 284.9 eV (curve iii) to 284.7 eV (curve v), consistent with the formation of C=C and C–C bonds in the branched oligomers.

Therefore, the XPS results support the reaction deduced in real space. **The XP spectra and their interpretation are added to the Supplementary Information (Supplementary Fig. 3) and referred to in the main text.**

2. The main drawback of the proposed mechanism is the lack of selectivity (see the dispersion of products in Fig. 4a). This is a known issue, particularly in C–H dehydrogenation reactions. Can the carbene-based oligomer formation be directed and induced in a more selective manner, perhaps through a more directional Ullmann coupling? This could improve the selectivity.

Reply: We thank the reviewer for the suggestion. While Ullmann coupling is indeed a powerful approach in on-surface synthesis, its drawback is the generation of halogen byproducts, which chemisorb on the surface, hinder the diffusion of reaction intermediates (Refs. 35, 36) and promote alternative reaction pathways (Ref. 37). In contrast, our carbene-based strategy is residue-free, as the N₂ byproduct desorbs at 87 K (Fig. R11c). It hence provides a complementary route to construct low-dimensional nanostructures on surfaces. We anticipate that the selectivity can be improved through rational precursor design and careful selection of the surface. **A comparison between our approach and Ullmann coupling has been included in the revised manuscript (pages 7,8).**

3. Similarly to query 1.2, have the authors considered the possibility that in structure “**3a**,” the final product might involve a surface metal atom rather than an adatom? DFT calculations could be helpful in addressing this.

Reply: We thank the reviewer for suggesting this structure. Following the suggestion, we have optimized two carbenes **2a** sharing one surface atom. However, this results in two individual carbenes **2a** adsorbed perpendicularly to the surface at neighboring surface sites. Such a structure would be imaged as two bright protrusions, similar to carbenes **2a** before annealing (Fig. 3b). It cannot explain the dimer-like **3a**, which is imaged as two pairs of three protrusions (Fig. 3d,e).

4. Why did the authors choose Ag? What about Au? It is likely that gold would result in more diffusion, less interaction, and a higher number of adatoms, potentially leading to more complex structures. A brief discussion of how the choice of surface affects the results would be valuable.

Reply:

Fig. R12. Adsorption of carbenes on coinage metal surfaces. Calculated adsorption energies and carbon-metal bond lengths for small carbenes on Cu(111), Ag(111), and Au(111) surfaces (adapted from Ref. 26).

We choose silver because silver-catalyzed carbene reactions are useful for various synthetic transformations, yet they are less explored compared to reactions catalyzed by gold (Ref. 25). Studying carbene reactions on well-defined single-crystal silver surfaces provides mechanistic insight into silver-catalyzed carbene reactions in solution. Moreover, recent DFT calculations for smaller carbenes reveal that carbenes interact more strongly with gold than with silver (Fig. R12). It is because the back-donation from the carbene to gold is stronger than that to silver at a similar donation from the metal to the carbene. Therefore, gold may not facilitate more diffusion even at a high number of adatoms to form more complex structures. **We have included a discussion of the choice of the surface in the revised manuscript (page 4).**

5. Making carbenes into oligomers is important, but doing so on a metal surface reduces the applicability of those oligomers. The article might be much more sound if the authors managed to perform the reaction on an insulating surface. This could work for “**2a**,” but not for the C–H dehydrogenation, as C–H cleavage typically requires a catalyst. However, the authors could evaporate "adatoms" to create a single-atom catalyst. This is just a speculative idea. Of course, this is not being requested for the manuscript publication.

Reply: We appreciate this suggestion. Carbene-based reactions on insulating surfaces have potential applications in nanoelectronics. As the reviewer notes, the main challenge of synthesis on insulating surfaces is the lack of catalytic activity. The photo-induced carbene formation does not necessarily require the catalytic effects of a metal surface. The subsequent C–H activation could, in principle, be catalyzed by foreign metal adatoms. In the present work, we focus on demonstrating the feasibility of carbene-mediated C–H activation on metal surfaces. Applying this concept to insulating surfaces is undoubtedly of interest for future studies. We have included an outlook in the revised manuscript discussing the potential applications of carbene reactions on insulating surfaces.

Reviewer #3

This is an elegant study on C-C coupling in carbene molecules, in the context of "on surface synthesis". Efforts in using surfaces to obtain planar conjugated structures started less than three decades ago and are becoming increasingly popular. This particular investigation focuses on obtaining highly branched oligomers.

The concept and results are original and the study was conducted methodically and rigorously. Typically the use of adjectives such as "novel" are discouraged, for fairly obvious reasons (novel platform... if this were not novel, we shouldn't even be discussing it).

Reply: We thank the reviewer for the very positive evaluation of our work. We agree that adjectives such as "novel" should be avoided, and we have removed them throughout the revised manuscript.

My main concerns are:

- the references. At this point, forty references for this kind of study is a bare minimum.

Reply: In the revised manuscript, we have increased the number of references to fifty-eight.

- the conclusions are too dry. I encourage the authors to describe a broader outlook, in the form of perspectives for future work.

Reply: We thank the reviewer for the suggestion. We have extended the conclusion to include a broader outlook: "Our proof-of-principle study establishes a platform for synthesizing structurally distinct branched oligomers on surfaces. The potential applications of this platform are promising. First, using diazo compounds as precursors of reactive carbenes offers a residue-free on-surface synthesis route that complements the widely used Ullmann coupling on surfaces, thus broadening the toolbox of on-surface synthesis. Second, this approach can be expanded to synthesize various N-doped 0D oligomers by replacing C2, C4, C5, or C7 atoms in carbene **2b** with N atoms. Moreover, the oligomer size can be tuned by tailoring the dimensions of the carbenes. Finally, this approach could be extended to synthesize low-dimensional nanostructures on nonmetallic surfaces. The photo-induced carbene formation does not necessarily require the catalytic effects of a metal surface. The subsequent carbene coupling could be initiated by co-adsorption of single Ag atoms. The platform thus paves the way to synthesize functional materials with applications in nanoelectronics."